# OBI CHARiot: Full-page OBI Rubbing Segmentation with Dual Data Flywheels

## Abstract

Oracle Bone Inscriptions (OBI), one of the earliest mature writing systems globally, serve as a crucial carrier of human civilization. However, directly segmenting OBI characters from full-page rubbings remains unexplored, primarily due to the scarcity of high-quality annotated data. To address this challenge, we propose a two-stage training framework, named `OBI CHARiot`, which establishes a cycle of mutual improvement between model performance and data quality. In the first stage, a data flywheel mechanism is employed to iteratively train a SAM2 while automatically aligning existing low-quality annotations, rather than directly utilizing the model's initial predictions. In the second stage, we employ an iterative strategy on a large collection of unlabeled rubbings, integrating automatic annotation with continuous model refinement. For reliable evaluation, we invite domain experts to annotate 2,226 rubbings, resulting a test set `OBIMDTest`. Experimental results demonstrate that OBI CHARiot offers advantages in both model performance and data quality. Specifically, training SAM2 with our framework yields a remarkable **14.99%** gain in mask $AP_{50}$ over the baseline trained on raw data. Similarly, various off-the-shelf instance segmentation models exhibit improved performance when trained on data processed by OBI CHARiot. Moreover, the characters segmented by our framework yield a **22.75%** improvement in top-1 accuracy on the downstream deciphering task. These findings confirm the significant potential of OBI CHARiot to advance OBI research. To support future studies, our model and the processed data will be made publicly available.

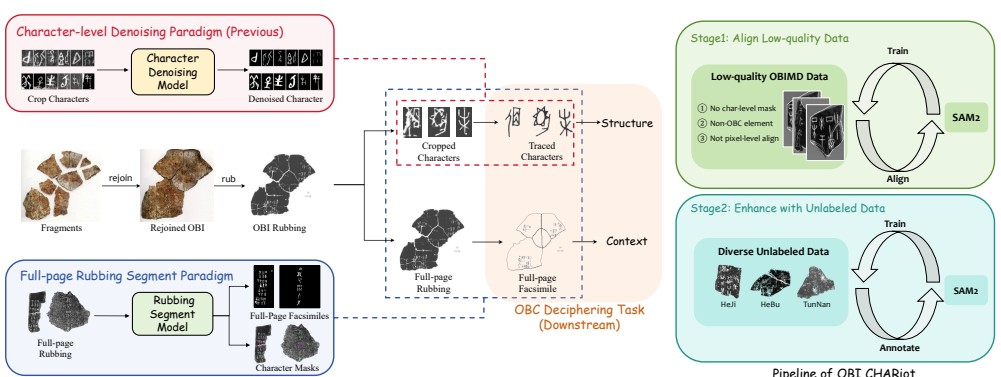

Figure 1: Left: We introduce a full-page rubbing segmentation paradigm to overcome the limitation of character-level denoising, which typically ignores the contextual information required for deciphering. Right: We propose a two-stage training framework based on the proposed paradigm, named OBI CHARiot, to address the issues of low-quality and scarcity in public datasets sequentially.

## 1 Introduction

Oracle Bone Inscriptions (OBI) are characters carved on turtle shells and animal bones during China's Shang Dynasty (around 1300 BCE). They are among the first known mature writing systems

not only in China but also globally (Meng, 2017; Liu et al., 2020; Fu et al., 2022; Fujikawa et al., 2023; Wang et al., 2024a). Deciphering Oracle Bone Characters (OBC), *i.e.* interpret the meaning of these ancient characters, serves as a core task in the field of OBI studies and has attracted significant interest from scholars (Chang et al., 2022; Guan et al., 2024). When deciphering OBC, scholars must not only rely on local glyph and structural features of characters but also integrate contextual information from full-page OBI rubbings to identify clues and bases for interpretation.

Due to the noise in OBI rubbings (*e.g.*, cracks, abrasions, and blurriness), it is often necessary to first trace OBI into clean facsimiles before deciphering. To date, the tracing process still relies heavily on manual work by experts. While several studies (Shi et al., 2022b;a; Wang et al., 2022) attempt to address this issue, mainly adopt a character-level denoising paradigm, *i.e.* denoise cropped regions, resulting in characters with clear glyph structure. Yet this paradigm focuses solely on character structures, overlooking the fact that deciphering OBC requires simultaneous consideration of contextual information to find evidence for interpretation, which demonstrate in Figure 1.

Given the limitations of the existing paradigm, we propose a new paradigm: `full-page rubbing segmentation`. Taking full-page rubbings as input, the paradigm aims to generate full-page facsimiles while independently segmenting an instance-level mask for each OBC. However, the only existing public dataset structured as OBI full-page rubbing-facsimile pairs is OBIMD (Li et al., 2024), yet its low quality and limited size make it difficult to directly train a robust full-page rubbing segmentation model. As shown in Figure 3, rubbings and facsimiles pairs are indeed not aligned at the pixel level, and the character masks are not annotated in instance-level in OBIMD. Furthermore, the dataset's scope is further limited by its composition, which draws only from a subset of JiaGuWenHeJi (HeJi) and discoveries at Huayuanzhuang Locus East within the Yin Ruins (HuaDong), resulting in constrained scale and variety.

To address the above issues, inspired by the data flywheel mechanism (Kirillov et al., 2023), *i.e.* co-develop the model with model-in-the-loop dataset annotation, we propose `OBI CHARiot`, a two-stage training framework built on the full-page rubbing segmentation paradigm, as illustrated in Figure 1. In the stage 1, the goal is to fully leverage existing low-quality annotated data. To this end, we first identify aligned OBC by comparing character regions in rubbings and facsimiles, yielding a small set of aligned data. We use this data to cold-start the SAM2 (Ravi et al., 2024), then further align the remaining OBC iteratively via a modified data flywheel: by aligning model predictions with ground-truth characters, we add aligned ground-truth characters rather than raw model predictions to the training set for subsequent iterations. Ultimately, after multiple iterations, we obtain high-quality aligned data. The stage 2 focuses on expanding data volume. We collect a large volume of unlabeled rubbings from the Internet; similarly, via a data flywheel, we automatically incorporate the model's high-confidence predictions into the training data, thereby expanding the data scale and further enhancing the model's performance. Additionally, to reliably evaluate the model's performance, we invite domain experts to annotate 2,226 rubbings, resulting in a high-quality test set named `OBIMDTest`.

Experimental results demonstrate that OBI CHARiot offers advantages in both model performance and data quality. Model trained with OBI CHARiot achieves a **9.52%** improvement in mIoU for facsimile generation and a **14.99%** increase in mask $AP_{50}$ for character-level segmentation. Furthermore, several off-the-shelf instance segmentation methods exhibit improved performance in full-page rubbing segmentation when trained on data processed by OBI CHARiot. Moreover, OBI CHARiot also achieves a **22.75%** improvement in top-1 accuracy for the OBC deciphering task, further demonstrating its utility in downstream tasks.

In summary, our contributions are as follows:

- Since the existing character-level denoising paradigm overlook the needs of contextual information for downstream deciphering tasks, we propose a new paradigm of full-page OBI rubbing segmentation.

- To address the issues of low-quality and scarcity in public datasets, we present OBI CHARiot, a two-stage training framework based on data flywheel mechanism. OBI CHARiot fully leverages existing low-quality ground-truth through iterative character alignment and expands the dataset volume by automatically annotating unlabeled rubbings.

- We invite experts to annotate a high quality test set OBIMDTest, for accurate evaluation. Experimental results on OBIMDTest demonstrate the dual superiority of our approach: not

only does OBI CHARiot itself achieve excellent performance, but the data it processes also consistently yields superior results across various instance segmentation models.

## 2 RELATED WORK

### 2.1 ORACLE BONE INSCRIPTION

OBI represent one of the earliest mature writing systems globally. In recent years, numerous studies (Qiao et al., 2024; Hu et al., 2024; Wu et al., 2025; Zhang et al., 2022; Hu et al., 2024) seek to leverage artificial intelligence (AI) technologies to facilitate the understanding of OBI. Among these efforts, deciphering OBC (*i.e.* interpreting the meaning of OBC) stands as a core task in the field of OBI research. It has also emerged as a major focus of AI-related investigations, spawning multiple distinct paradigms. Diao et al. (2023b;a) frame deciphering as character recognition task and achieve zero-shot recognition by decomposing OBC into radicals. Meanwhile, Sundial-GAN (Chang et al., 2022) and OBSD (Guan et al., 2024) directly transform OBC into modern Chinese characters via generative models. Additionally, OBI-Bench (Chen et al., 2024) and V-Oracle (Qiao et al., 2025) attempt to leverage the capabilities of vision-language models to directly explain the meaning of OBC.

However, these deciphering approaches still heavily rely on hand-print characters traced by human experts. To reduce the dependency on character-level hand-annotated data, researchers have developed various approaches (Jiang et al., 2023; Zhang et al., 2024b; Diao et al., 2025; Li et al., 2025a) aimed at converting character regions from OBI rubbings into clean character images that closely resemble expert-traced facsimiles. STSN (Wang et al., 2022) achieves mutual conversion between facsimile and rubbing images by disentangling glyph information from noise. CharFormer (Shi et al., 2022a) and RCRN (Shi et al., 2022b) leverage the skeleton structures of characters to preserve their inherent structural integrity during the denoising process. Li et al. (2025b), on the other hand, construct a structure-aligned, expert-annotated dataset specifically for OBC denoising and adapt a diffusion model to generate high-quality clean OBI characters. Despite these progress, they are constrained by the character-level denoising paradigm, as they fail to account for the need to decipher OBC within the context. To tackle this limitation, we propose a new paradigm that processes full-page OBI rubbings to generate both full-page facsimiles and character masks, preserving both character-level integrity and page-level context.

### 2.2 DATA ENGINE AND DATA FLYWHEELS

As computer vision models mature, the scarcity and limited quality of labeled data have increasingly become a bottleneck for further performance improvements. The traditional paradigm, which heavily relies on manual annotation, struggles to generate large-scale training data, which has led to the emergence of data engines or data flywheels (Kirillov et al., 2023; Ravi et al., 2024). A data flywheel enables low-cost enhancements to data quality and scale through automated sample selection, annotation, and mining processes, and has been widely adopted across various domains (Liang et al., 2024; Wang et al., 2024b; Pan et al., 2025; Wei et al., 2024; Zhou et al., 2024). Specifically, Depth Anything (Yang et al., 2024) builds a pseudo-labeling system for depth estimation; RAM (Zhang et al., 2024a) introduces a new paradigm for image tagging, leveraging large-scale image-text pairs processed by a data engine; and Xiao et al. (2024) constructs the large-scale, high-quality FLD-5B dataset using an iterative automated image annotation strategy—while simultaneously developing the powerful vision foundation model Florence-2. In this space, SAM (Kirillov et al., 2023) stands as a representative example of data flywheel implementation. Initially, SAM's training relied on manual annotations; after accumulating a certain volume of data, it only required manual labeling of specific hard-to-annotate objects. Eventually, it could significantly expand the dataset based on the model's prediction confidence.

However, for our rubbing segmentation, data misalignment results in poor quality, making it difficult to directly adapt the data flywheel mechanism used in SAM. To address this issue, we developed OBI CHARiot: a novel framework centered on pixel-level alignment. Our framework operates by iteratively training the model on progressively aligned data, which is identified by comparing the model's output with ground truth annotations, and ultimately yields a high-quality dataset.

# 3 OBI CHARIOT

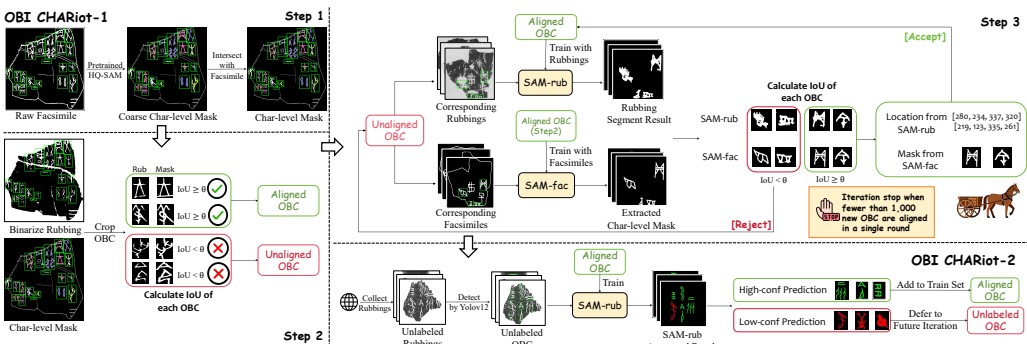

Figure 2: The pipeline of OBI CHARiot.

This section first introduces the data issues in OBIMD (Section 3.1). It then details our two-stage OBI CHARiot framework, as illustrated in Figure 2: Stage 1 (OBI CHARiot-1) leverages low-quality OBIMD annotations to train a segmentation model and build a pixel-aligned dataset (Section 3.2). To address data scarcity, Stage 2 (OBI CHARiot-2) employs a data flywheel on unannotated web rubbings for model-in-the-loop annotation, as described in Section 3.3.

## 3.1 DATA ISSUES IN OBIMD

Our task is conducted on the OBIMD dataset (Li et al., 2024), which is currently the only public resource containing paired full-page OBI rubbings and expert-traced facsimiles. However, the dataset suffers from critical issues of low quality, as shown in Figure 3. Specifically, the expert-traced facsimiles in OBIMD, originally intended for documentary purposes, are **seldom pixel-aligned** with their corresponding rubbings. Furthermore, these facsimiles are single-channel binary images that **lack individual character masks** and **include various non-character elements** (*e.g.*, boundary lines and rejoining lines). In addition to these quality limitations, OBIMD is also restricted in scale and diversity. It comprises only

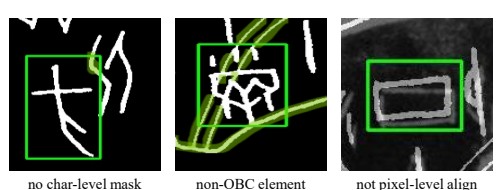

no char-level mask    non-OBC element    not pixel-level align

Figure 3: OBIMD suffer from low-quality: they lack independent character masks, and directly extracting masks from boxes may include adjacent characters or non-character elements. Furthermore, the facsimiles may not be pixel-aligned with the rubbings, and this misalignment can introduce noise during training.

about one-fifth of the rubbings recorded in HeJi and a few hundred rubbings unearthed at HuaDong. These constraints collectively make it impractical to train a reliable full-page rubbing segmentation model directly on this dataset.

## 3.2 OBI CHARIOT-1: ALIGN LOW-QUALITY DATA

The goal of the OBI CHARiot-1 is to leverage low-quality datasets to train a high-performance rubbing segmentation model while simultaneously deriving a high-quality dataset. OBI CHARiot-1 comprises three key steps. First, we utilize a pre-trained HQ-SAM (Ke et al., 2023) to extract character-level masks. Second, we identify pixel-aligned OBC pairs from the raw dataset by comparing the binarized rubbings with the extracted character masks. This initial dataset is then used to cold-start a SAM2. Subsequently, a modified data flywheel mechanism (Kirillov et al., 2023) is employed to iteratively align the remaining OBC through joint model-data optimization.

**Step 1: Extract char-level masks.** We use the ground-truth bounding boxes as prompts and input full-page binary facsimiles into the pre-trained HQ-SAM to extract character-level masks. Given that the masks from the pre-trained HQ-SAM may not accurately recover OBC glyph, we refine

them by taking a pixel-wise intersection with the corresponding regions of the raw facsimiles. This operation preserves the glyph structures (*e.g.*, hollow regions) from the facsimiles while leveraging the HQ-SAM outputs to filter out non-character elements.

**Step 2: Identify aligned characters.** Leveraging the coarse character-level masks obtained in the previous step, we next identify pixel-aligned characters from the raw dataset. This is done by calculating the intersection over union (IoU) between each character mask and the corresponding region in the binarized full-page rubbing. A character is considered aligned if its IoU exceeds $\theta$.

**Step 3: Iteratively align remaining characters.** At this step, we adopt a modified data flywheel approach to iteratively align the remaining data and enhance model performance. We first train a SAM2 (referred to as SAM-rub) using the aligned characters from previous step, with full-page rubbings as input. Additionally, since the character-level masks extracted by the pre-trained HQ-SAM in Step 1 are relatively coarse, we proceed to train a new SAM2 (referred to as SAM-fac). SAM-fac is trained using the ground-truth facsimiles as input and the character-level masks as targets. To enable both SAM-rub and SAM-fac to output more precise structural features, we append lightweight deconvolution layers to each model, upscaling their output masks resolution from $256 \times 256$ to $1024 \times 1024$.

Next, we implement the data flywheel approach, where each iteration consists of two sequential steps: **an alignment step and a training step**. In the alignment step, we align the remaining OBC by integrating the prediction from SAM-rub with the character mask extracted by SAM-fac. The process begins by using the ground-truth bounding boxes as prompts for both models: the rubbing is fed into SAM-rub to locate the character, while the corresponding facsimile is fed into SAM-fac to extract an accurate mask. The foreground regions from both outputs are cropped and compared. If their IoU exceeds $\theta$, the character is consider accepted. Otherwise, the alignment is rejected and deferred to a future iteration. To account for potential positional misalign between the raw facsimile and the rubbing, the accepted SAM-fac mask is then repositioned to the location identified by SAM-rub. This automatically aligned mask is subsequently added to the training dataset for the next iteration. In the subsequent training step, we train a new SAM-rub using the updated aligned OBC and incorporating the newly trained SAM-rub in the following alignment step.

Iterations proceed until fewer than 1,000 new OBC are aligned in a single alignment step. For the remaining cases, we crop the foreground regions from the SAM-rub and SAM-fac output masks and apply slight translational shifts. The position yielding the highest IoU is selected as the final mask, ultimately producing a complete, high-quality dataset for full-page rubbing segmentation. This final dataset is then used to train the final version of SAM-rub, resulting in a high-performance rubbing segmentation model.

### 3.3 OBI CHARIOT-2: ENHANCE WITH UNLABELED DATA

Given the limited diversity and quantity of rubbings in the OBIMD dataset, we collect a large number of unlabeled rubbings from the Internet to expand the existing dataset. After deduplicating against the OBIMD dataset by source, we additionally gathered a total of 52,311 rubbings, including 32,965 contained in HeJi, 14,737 in JiaGuWenHeJiBuBian (HeBu), and 4,609 unearthed at Xiaotun South Locus (TunNan). This new collection covers more types of rubbings and excavation sites, significantly enriching the training data pool.

In OBI CHARiot-2, we adopt a data flywheel approach to iteratively achieve joint model-data optimization. Different from OBI CHARiot-1, where we can use ground-truth boxes as prompts, we have no annotation available here. Although SAM2 can segment instances using grid point prompts, the sparse and noisy nature of OBC foreground pixels makes the result prone to over-segmentation and missed detection. Therefore, we train a YOLOv12 (Tian et al., 2025) based on our aligned OBIMD dataset to detect OBC in unlabeled full-page rubbings, and use the predict boxes as prompts for SAM-rub. A comparison of the performance using grid points versus YOLO boxes as prompts is presented in Section 4.6.

Each iteration of the data flywheel comprises **an annotate step and a training step**. In the annotate step, SAM-rub uses the predicted boxes as prompts to generate character-level masks for unannotated rubbings. Masks with high confidence are then added to the training data. In the training step, SAM-rub is first pre-trained on newly annotated data and then fine-tuned on the data aligned by OBI

CHARiot-1. This iterative process continues until fewer than 5,000 new OBC are aligned in a single annotate step.

# 4 EXPERIMENTS

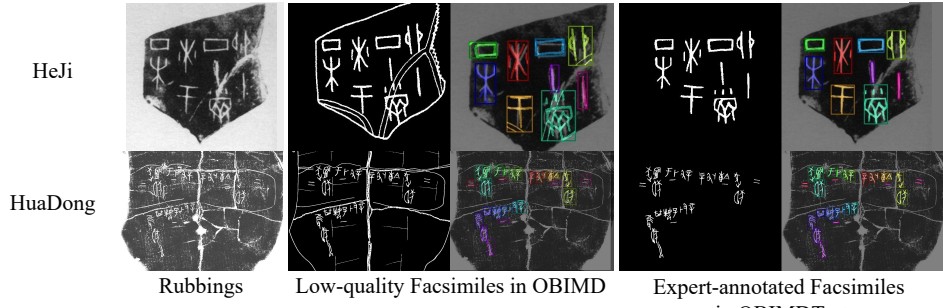

HeJi

HuaDong

Rubbings     Low-quality Facsimiles in OBIMD     Expert-annotated Facsimiles in OBIMDTest

Figure 4: Data from HeJi and HuaDong, incorporating full-page rubbings, along with the raw facsimiles from OBIMD and the expert-annotated facsimiles from our OBIMDTest.

Table 1: Rubbing segmentation performance of SAM2 trained by different frameworks on OBIMDTest-HJ. The optimal results are highlighted in **bold**, the suboptimal results are underlined, and symbol $\dagger$ indicates the use of additional unlabeled data.

| Framework | Character Segmentation | | Facsimile Generation | |
|---|---|---|---|---|
| | mask $AP_{50}$ | mask AP | mIoU | F-score |
| Raw Data | 76.39 | 21.98 | 58.65 | 73.68 |
| Char Denoise$_{SegFormer}$ | 88.27 | 32.85 | 65.59 | 79.07 |
| Char Denoise$_{CharFormer}$ | 90.16 | 36.61 | 66.70 | 79.86 |
| SAM-style Training | 87.70 | 33.98 | 68.15 | 80.88 |
| OBI CHARiot-1 | **91.38** | 39.46 | 68.17 | 80.95 |
| OBI CHARiot-2$^{\dagger}$ | 91.33 | **39.67** | **68.38** | **81.09** |

## 4.1 SETTINGS

**Data Statistic**: We conduct experiments on the OBIMD dataset (Li et al., 2024), which contains 9,913 rubbings from HeJi and 164 rubbings unearthed at HuaDong. We randomly divide the rubbings from HeJi into a training set of 7,850 images and a test set of 2,063 images, while the entire set of HuaDong rubbings is reserved exclusively for Out-of-Distribution (OOD) performance evaluation.

**Test Set Construction**: To address the quality limitations of the OBIMD dataset (see Figure 3), we constructed a high-quality test set through an expert-driven annotation pipeline. This process involved two key steps: first, several graduate students specializing in OBI align the OBC based on the raw facsimiles from OBIMD; subsequently, two OBI experts verify the annotations to ensure correct glyph structures and consistency with the OBC on the corresponding rubbings. The outcome is the OBIMDTest dataset, comprising two subsets: OBIMDTest-HJ (derived from the HeJi source) and OBIMDTest-HD (derived from the HuaDong unearthed rubbings). Sample data from OBIMDTest are shown in Figure 4.

**Compared Training Frameworks**: Our goal is to validate two claims. First, SAM2 (Ravi et al., 2024) trained by our OBI CHARiot exhibit superior full-page rubbing segmentation performance compared to SAM2 trained via alternative frameworks. Second, the data processed through our OBI CHARiot features higher quality. Specifically, when the same instance segmentation model is deployed, it achieves more robust segmentation results on data processed by OBI CHARiot. To verify these claims, we compared the following training frameworks:

- **Raw Data**: SAM2 is trained directly on the original OBIMD data, where character masks are extracted by applying the bounding boxes to the facsimiles.

- **Char Denoise**: Building upon the character-level denoising paradigm, we employ a semantic segmentation model, SegFormer (Xie et al., 2021), and a character denoising model, CharFormer (Shi et al., 2022a), to denoise cropped character regions separately. We then train SAM2 using the denoised data generated by these two models, respectively.

- **SAM-style Training**: This framework adopts the data flywheels of SAM (Kirillov et al., 2023). Specifically, after cold-start a SAM2 using aligned characters in step 2 (see Section 3.2), high-confidence predictions are directly incorporated into the subsequent training dataset, without validating whether these predictions align with ground truth annotations.

- **OBI CHARiot-1** and **OBI CHARiot-2**: SAM2 trained by our OBI CHARiot framework. IoU threshold is set to $\theta = 0.6$ in our experiments.

**Evaluation**: We use mask $AP_{50}$ and mask AP to evaluate the quality of **Character Segmentation**, and employ mIoU and F-score to assess the quality of **Facsimile Generation**, respectively. For all experiments, we generate the full-page facsimile for each rubbing by merging its corresponding predicted character masks.

## 4.2 SEGMENTATION CAPABILITIES OF MODEL

Table 2: Rubbing segmentation performance of instance segmentation models on OBIMDTest-HJ under data processed by training frameworks.

| Model | Data Source | Character Segmentation | | Facsimile Generation | |
|---|---|---|---|---|---|
| | | mask $AP_{50}$ | mask AP | mIoU | F-score |
| Mask R-CNN | Raw data | 59.63 | 13.30 | 50.10 | 66.31 |
| | Char Denoise$_{SegFormer}$ | 71.80 | 21.11 | 57.52 | 72.52 |
| | Char Denoise$_{CharFormer}$ | 73.56 | 24.26 | 59.46 | 74.03 |
| | SAM-style Training | 72.99 | 24.00 | 57.98 | 72.81 |
| | OBI CHARiot-1 | 77.84 | 23.90 | 59.77 | 74.37 |
| | OBI CHARiot-2$^{\dagger}$ | **83.14** | **30.46** | **62.55** | **76.65** |
| YOLACT | Raw data | 55.54 | 12.69 | 49.83 | 65.71 |
| | Char Denoise$_{SegFormer}$ | 73.68 | 24.35 | 59.57 | 74.09 |
| | Char Denoise$_{CharFormer}$ | 73.88 | 25.99 | 60.05 | 74.50 |
| | SAM-style Training | 69.32 | 21.81 | 55.96 | 71.05 |
| | OBI CHARiot-1 | 74.92 | 25.75 | 60.05 | 74.49 |
| | OBI CHARiot-2$^{\dagger}$ | **78.74** | **29.81** | **62.44** | **76.45** |
| SOLO | Raw data | 41.76 | 8.85 | 48.83 | 65.05 |
| | Char Denoise$_{SegFormer}$ | 65.38 | 18.69 | 56.30 | 71.45 |
| | Char Denoise$_{CharFormer}$ | 67.65 | 21.84 | 58.62 | 73.30 |
| | SAM-style Training | 67.38 | 19.58 | 58.77 | 73.60 |
| | OBI CHARiot-1 | 70.04 | 22.00 | 58.43 | 73.28 |
| | OBI CHARiot-2$^{\dagger}$ | **79.15** | **26.93** | **61.61** | **75.84** |
| Mask2Former | Raw data | 50.68 | 11.20 | 50.46 | 66.71 |
| | Char Denoise$_{SegFormer}$ | 72.67 | 23.56 | 60.22 | 74.71 |
| | Char Denoise$_{CharFormer}$ | 73.93 | 24.70 | 61.18 | 75.55 |
| | SAM-style Training | 70.03 | 22.18 | 57.78 | 72.79 |
| | OBI CHARiot-1 | 74.28 | 24.20 | 61.17 | 75.55 |
| | OBI CHARiot-2$^{\dagger}$ | **80.61** | **27.92** | **62.43** | **76.53** |

To evaluate the segmentation performance of our OBI CHARiot-trained SAM2, we benchmark it against SAM2 models trained under different frameworks on the OBIMDTest-HJ, with ground-truth bounding box as prompt, as shown in the Table 1. Training with raw data yield poor results because the data contained non-character noise and is not pixel-level aligned. In contrast, other frameworks mitigate noise and biases in the data, achieving improved segmentation performance.

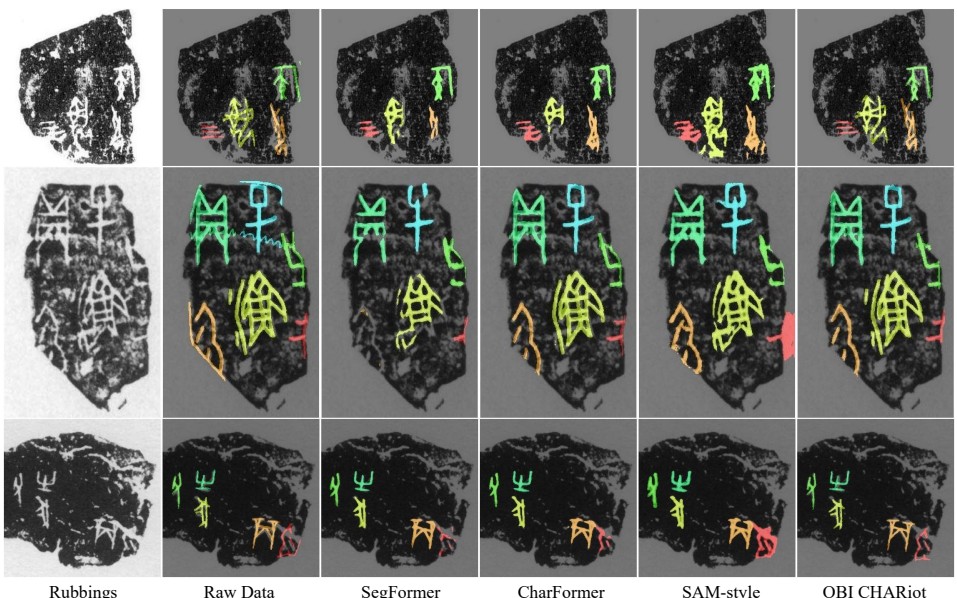

| Rubbings | Raw Data | SegFormer | CharFormer | SAM-style | OBI CHARiot |

Figure 5: Visualizations of training data processed by different framework.

The performance of Char Denoise on facsimile generation is limited because its character denoising model is trained on low-noise characters, which fails to handle some challenging cases. Similarly, SAM-style Training underperforms in character segmentation since it incorporates its own high-confidence predictions into training, while these predictions may contain inferior glyph structures compared to the OBIMD.

In contrast, OBI CHARiot achieves superior performance by introducing an alignment-based data flywheel. Instead of relying on potentially flawed high-confidence outputs, it enhances the training set with masks extracted from expert-traced facsimiles using a dedicated SAM-fac model. This strategy allows for more effective utilization of the precise glyph structures available in the ground-truth annotations.

## 4.3 QUALITY OF PROCESSED DATA

To validate the quality of data processed by different frameworks, we conduct experiments on several off-the-shelf instance segmentation models with diverse architectural designs. The evaluated models include: CNN-based methods (He et al., 2016) such as Mask R-CNN (He et al., 2017) (two-stage), YOLACT (Bolya et al., 2019) (single-stage), and SOLO (Wang et al., 2020) (box-free); along with the Transformer-based (Liu et al., 2021) Mask2Former (Cheng et al., 2022). As shown in the Table 2, the raw OBIMD data yield limited model performance due to its poor quality. Notably, models train on data processed by OBI CHARiot-1 outperforming the baseline by an average of 22.37% in mask $AP_{50}$. As visualized in Figure 5, compared with data from other training frameworks, the data from our OBI CHARiot exhibits superior alignment and enhanced preservation of key OBC structures, such as holes and high-noise regions (*e.g.*, yellow OBC in the top row and red OBC in the bottom row). This higher data quality directly enables segmentation models to attain enhanced capabilities. Moreover, using data annotated in OBI CHARiot-2, models across all architectures achieve their peak performance: mask $AP_{50}$ improves by 6.14%, underscoring the value of the expanded dataset. Additional visualization results are provided in Appendix B.

## 4.4 OUT-OF-DISTRIBUTION PERFORMANCE ON OBIMDTEST-HD

We hold out data in OBIMD sourced HuaDong in training and evaluate the OOD performance of different frameworks on the OBIMDTest-HD dataset. It should be noted that the additional rubbings collected in Section 3.3 are also subjected to deduplication against the HuaDong data. As shown in Figure 4, compared with HeJi, the characters on HuaDong rubbings are more densely distributed,

Table 3: OOD rubbing segmentation performance on OBIMDTest-HD. The content in parentheses in the mask $AP_{50}$ column indicates the performance degradation of OBIMDTest-HD compared to OBIMDTest-HJ.

| Framework | Character Segmentation | | Facsimile Generation | |
| --- | --- | --- | --- | --- |
| | mask $AP_{50}$ | mask AP | mIoU | F-score |
| Raw Data | 35.88 (-40.51) | 5.70 | 49.75 | 66.36 |
| Char Denoise$_{SegFormer}$ | 45.98 (-43.98) | 8.11 | 50.56 | 67.06 |
| Char Denoise$_{CharFormer}$ | 40.81 (-49.35) | 7.54 | 50.04 | 66.61 |
| SAM-style Training | 52.00 (-35.70) | 8.84 | 51.84 | 68.19 |
| OBI CHARiot-1 | 61.17 (-30.21) | 11.85 | 52.71 | 68.94 |
| OBI CHARiot-2[†] | **63.20 (-28.13)** | **12.48** | **52.98** | **69.18** |

accompanied by larger cracks and more severe abrasion, these factors make character segmentation significantly more challenging.

As experimental results shown in Table 3, all frameworks exhibit a notable decline. However, OBI CHARiot still demonstrates stronger capabilities comparing to other training frameworks. Additionally, by integrating extensive unannotated rubbings for data expansion, OBI CHARiot-2 exhibits superior zero-shot capability compared to OBI CHARiot-1, achieves a 2.03% improvement in mask $AP_{50}$. Furthermore, compared with other methods, OBI CHARiot shows a smaller drop in mask $AP_{50}$ on OBIMDTest-HD, indicating that models trained via OBI CHARiot framework have advantages in terms of robustness. Further analysis is provided in Appendix C.

## 4.5 CHARACTER SIMILARITY AND DECIPHERING PERFORMANCE

Table 4: Similarity and deciphering accuracy of character segmented by difference training framework.

| Framework | Similarity | | ResNet | | EfficientNet | | ViT | | Swin | |
| --- | --- | --- | --- | --- | --- | --- | --- | --- | --- | --- |
| | SSIM | PSNR | Top-1 | Top-5 | Top-1 | Top-5 | Top-1 | Top-5 | Top-1 | Top-5 |
| Raw data | 0.5458 | 7.54 | 50.01 | 69.32 | 53.35 | 68.59 | 54.05 | 67.20 | 54.97 | 72.01 |
| Char Denoise$_{SegFormer}$ | 0.5972 | 8.87 | 65.55 | 83.01 | 69.67 | 82.47 | 69.84 | 81.04 | 71.37 | 85.53 |
| Char Denoise$_{CharFormer}$ | 0.6082 | 9.24 | 69.41 | 85.55 | 72.72 | 85.54 | 72.51 | 83.75 | 73.81 | 87.49 |
| SAM-style Training | 0.6114 | 9.31 | 71.95 | 87.74 | 75.46 | 87.16 | 75.60 | 85.73 | 76.67 | 89.57 |
| OBI CHARiot-1 | 0.6162 | 9.39 | 73.05 | 88.21 | 76.49 | 87.99 | 76.60 | 86.62 | 77.24 | 90.01 |
| OBI CHARiot-2[†] | **0.6207** | **9.50** | **73.91** | **88.66** | **77.36** | **88.89** | **77.69** | **87.33** | **77.87** | **90.05** |

To validate that the characters segmented by OBI CHARiot exhibit superior glyph structures, we first assess their image similarity to the ground-truth on the OBIMDTest-HJ. Furthermore, we conduct a deciphering task on OBIMD to verify the practical advantages of these improved structures. Specifically, we formulate the OBC deciphering task as a classification problem (Diao et al., 2023a;b). We train various classifiers (ResNet (He et al., 2016), EfficientNet (Tan & Le, 2019), ViT (Dosovitskiy, 2020), Swin (Liu et al., 2021)) on OBIMD, using their Top-1/Top-5 accuracy to evaluate structural quality.

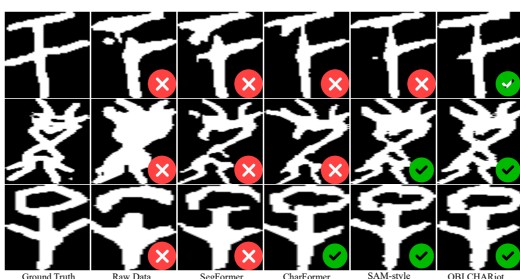

Figure 6: OBC segmented by different framework and their deciphering result evaluated by ResNet.

As shown in Table 4, the characters segmented by our OBI CHARiot framework achieve higher similarity scores and deciphering accuracy. Qualitative results in Figure 6 further support these findings. Compared to other methods, OBI CHARiot produces segmentation results with less noise and captures complex topological structures (*e.g.*, holes) more accurately, which contributes to more reliable decipherment.

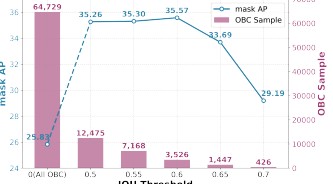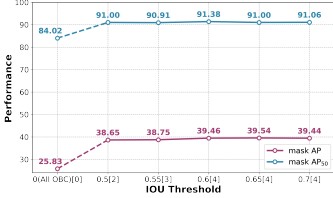

(a) The quantity of OBC and the performance of the cold-start SAM2 at different IoU thresholds in step 2.

(b) The performance of SAM2 after step 3.

Figure 7: The character segmentation performance of the SAM2 at different IoU thresholds. The numbers in [] in (b) indicate the number of iterations in Step 3.

## 4.6 ABLATION STUDY

**Is OBI CHARiot sensitive to IoU threshold?** In OBI CHARiot-1, we determine whether to accept data based on whether IoU$\geq \theta$. In Step 2, we need to identify a small number of high-quality aligned OBC in OBIMD and use this subset to cold-start a SAM2, enabling it to acquire preliminary segmentation capability. The results in Figure 7a show that when setting IoU=0.6, we can achieve certain segmentation capability with a relatively small amount of data. If all OBC extracted in Step 1 are used for training, the inclusion of low-quality characters leads to significantly worse overall performance. In Step 3, we incorporate the well-aligned outputs from SAM-rub and SAM-fac into the training set during the alignment step. As shown in Figure 7b, OBI CHARiot-1 is robust to the choice of threshold.

**Does using YOLO boxes as prompts in OBI CHARiot offer advantages over grid points?** As described in Section 3.3, we train YOLOv12 (Tian et al., 2025) use YOLO boxes as prompts instead of grid points in automatic annotation. The model achieves a box AP of 67.33 and a box AP$_{50}$ of 93.12. As shown in Table 5, using YOLO boxes as prompts yields higher segmentation performance compared to using grid points (a $32 \times 32$ grid). Figure 8 further demonstrate that the grid-point approach is more susceptible to issues such as over-segmentation and missed detections. Although a performance gap remains between using YOLO boxes and the ground-truth boxes, the experiments in Section 4.3 show that OBI CHARiot-2, which utilizes YOLO for data annotation, provides substantial benefits for training various instance segmentation models.

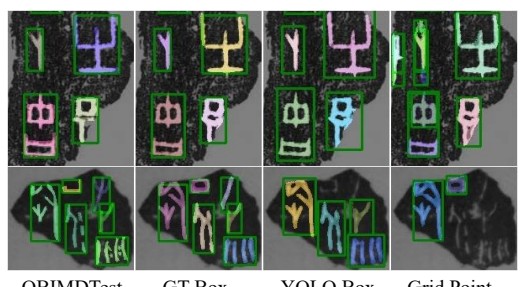

| OBIMDTest | GT Box | YOLO Box | Grid Point |

Figure 8: Visualization of facsimiles in OBIMDTest and OBI CHARiot-1 inference with different prompt.

Table 5: Performance of OBI CHARiot-1 with different prompts on the OBIMDTest-HJ.

| Prompt | mask AP$_{50}$ | mask AP |
|---|---|---|
| GT boxes | 91.38 | 39.46 |
| YOLO boxes | 73.78 | 30.13 |
| grid points | 63.16 | 23.65 |

## 5 CONCLUSION

This paper addresses the limitations of existing character-level denoising paradigms, which often overlook contextual information crucial for downstream OBC deciphering tasks. To overcome this, we propose full-page rubbing segmentation. However, directly training a robust model for this task is challenging due to data low quality and scarcity. Thus, we introduce OBI CHARiot, a training framework that implements a two-stage approach based on a data flywheel mechanism. Experimental results demonstrate that OBI CHARiot achieves superior model performance and higher data quality.

ETHICS STATEMENT

This work adheres to the ICLR Code of Ethics. In this study, no human subjects or animal experimentation was involved. All datasets used, including OBIMD (Li et al., 2024), were sourced in compliance with relevant usage guidelines, ensuring no violation of privacy. We have taken care to avoid any biases or discriminatory outcomes in our research process. No personally identifiable information was used, and no experiments were conducted that could raise privacy or security concerns. We are committed to maintaining transparency and integrity throughout the research process.

REPRODUCIBILITY STATEMENT

We have made every effort to ensure that the results presented in this paper are reproducible. All code and datasets have been made publicly available in an anonymous repository to facilitate replication and verification. The experimental setup, including training steps, model configurations, and hardware details, is described in detail in the paper. We have also provided a full description of OBI CHARiot to assist others in reproducing our experiments.

Additionally, the dataset we used (*i.e.* OBIMD) are publicly available, ensuring consistent and reproducible evaluation results.

We believe these measures will enable other researchers to reproduce our work and further advance the field.

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

## A    LLM Usage

Large Language Models (LLMs) were used to aid in the writing and polishing of the manuscript. Specifically, we used an LLM to assist in refining the language, improving readability, and ensuring clarity in various sections of the paper. The model helped with tasks such as sentence rephrasing, grammar checking, and enhancing the overall flow of the text.

It is important to note that the LLM was not involved in the ideation, research methodology, or experimental design. All research concepts, ideas, and analyses were developed and conducted by the authors. The contributions of the LLM were solely focused on improving the linguistic quality of the paper, with no involvement in the scientific content or data analysis.

The authors take full responsibility for the content of the manuscript, including any text generated or polished by the LLM. We have ensured that the LLM-generated text adheres to ethical guidelines and does not contribute to plagiarism or scientific misconduct.

## B    More Visualization Result

We provide additional visualization results below, where approximately 10% of OBC in both Seg-Former (Xie et al., 2021) and CharFormer (Shi et al., 2022a) cannot be confidently predicted, and the corresponding characters still retain the original annotations from OBIMD (Li et al., 2024).

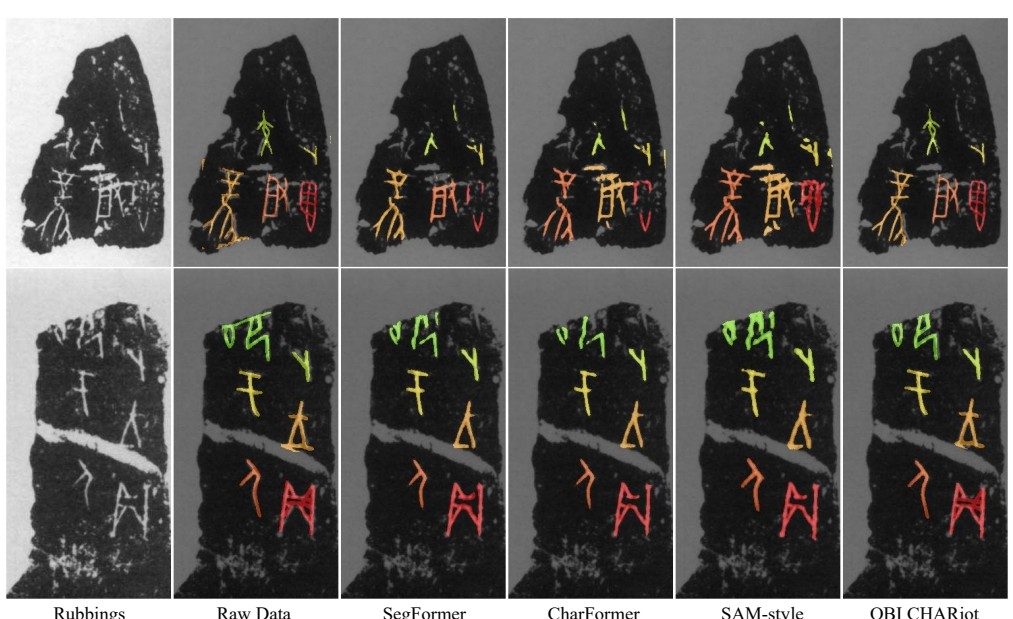

| Rubbings | Raw Data | SegFormer | CharFormer | SAM-style | OBI CHARiot |

Figure 9: Visualizations of training data processed by different framework.

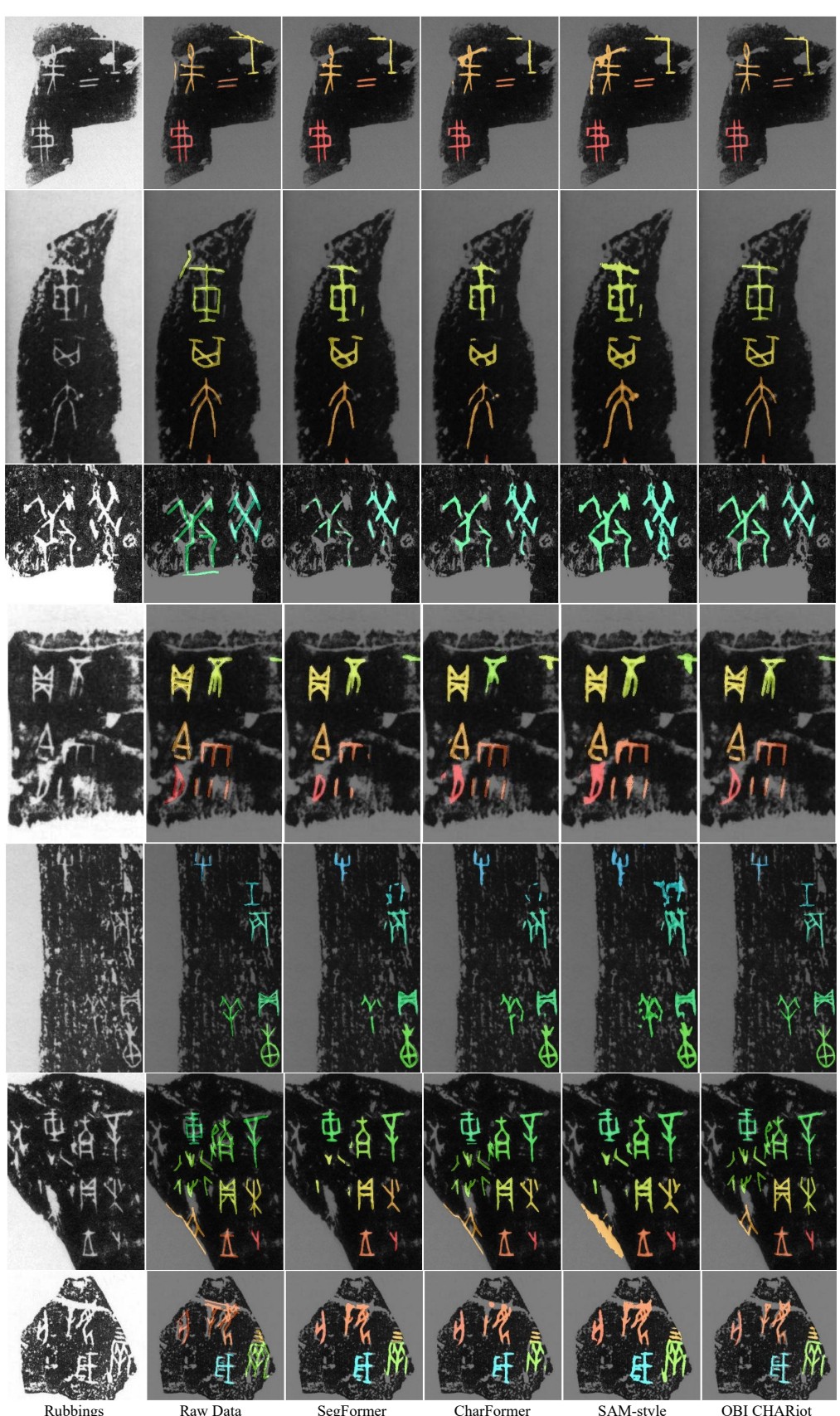

Figure 10: Visualizations of training data processed by different framework.

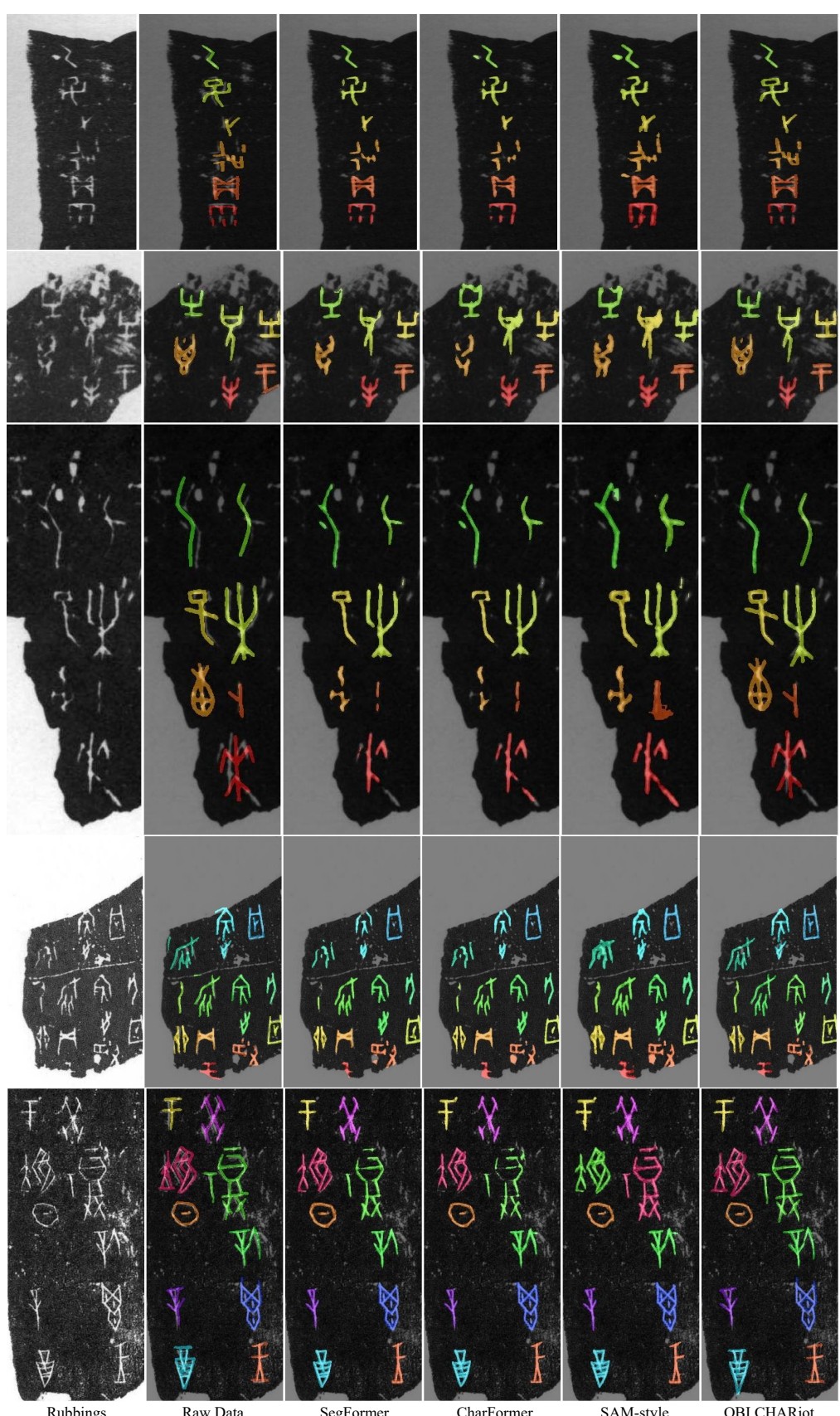

Figure 11: Visualizations of training data processed by different framework.

## C ANALYSIS OF PERFORMANCE ON OBIMDTEST-HD

Visualization results on the OBIMDTest-HD dataset are shown in Figure 12. As described in Section 4.4, the data in the HuaDong subset contains large areas of high noise, characterized by larger cracks and more severe abrasion compared to HeJi, which leads to its degraded performance. In contrast to other frameworks, OBI CHARiot is capable of producing better character shape predictions in these high-noise regions. It successfully avoids either misclassifying all cracked areas as foreground or simply discarding them.

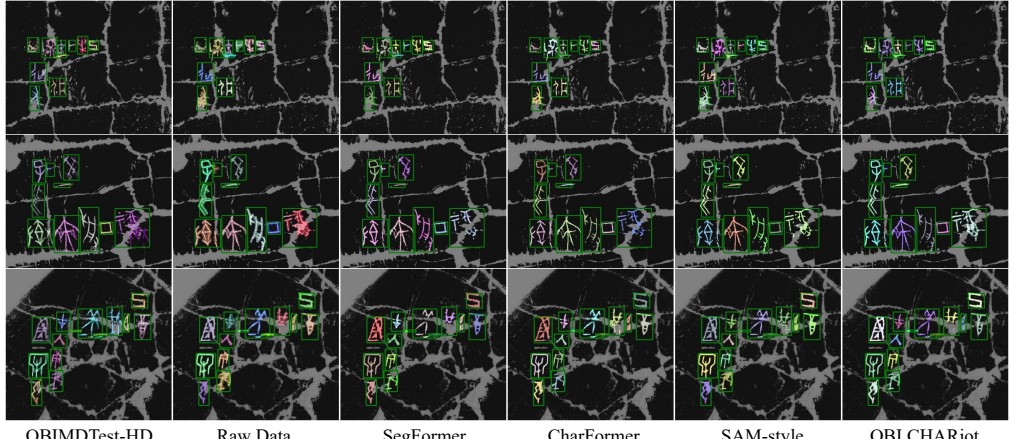

| OBIMDTest-HD | Raw Data | SegFormer | CharFormer | SAM-style | OBI CHARiot |

Figure 12: Visualizations of ground-truth and prediction of SAM2 trained on different framework.

