# OpenReview forum: "OBI CHARiot: Full-page OBI Rubbing Segmentation with Dual Data Flywheels"
_ICLR.cc/2026/Conference — Submitted to ICLR 2026_

### Official Review · Reviewer_Lf6A · 2025-10-20

**Soundness:** 2
**Presentation:** 2
**Contribution:** 2
**Rating:** 4
**Confidence:** 5

**Summary:**

This article proposes OBI CHARiot, a segmentation framework based on SAM for the oracle bone inscriptions on full-page rubbings, which follows a two-stage data collection-training strategy. A high-quality test set termed OBIMDTest containing 2,226 rubbings is built. Experiments show that models trained via OBI CHARiot outperform those trained with other frameworks. Meanwhile, OBI CHARiot achieves significant improvement in top-1 accuracy for the downstream OBI deciphering task.

**Strengths:**

1. Reasonable motivations from character-level denoising to full page-level segmentation task.
2. An semiautomatic framework to iteratively train and refine the model.
3. The experimental results have shown a decent improvement.

**Weaknesses:**

1. The technical contributions are relatively limited mostly related to engineering implementation issues. The core of OBI CHARIOT focuses on building the pipeline using SAM. The replaceability of this part is worth discussing to highlight the generality of this framework. Meanwhile, the contribution of the dataset part mainly focused on the modification of OBIMD, and the new contributions were relatively limited.
2. Lack of some details, such as the prompt information for SAM model at different stages of *OBI CHARIOT.*
3. Some illustrations could be more clear, such as the pixel union operation between the outputs of SAM and raw binary facsimiles (add in Fig .2).
4. I suggest that the illustration in Fig. 2 should connect OBI CHARiot-1 and OBI CHARiot-2. Currently, it is not clear enough, considering the numerous steps mentioned.
5. Missing detailed information about the personnel with expert annotations.
6. The so-called decipherment in the experiment is actually a classification task. This point must not be confused.

**Questions:**

1. Some writings and typos should be fixed, such as “bulit” in line 79, the necessary space after "i.e.” in line 183, capitalization issue of “Training’s” in line 361, “*prmopt*” in line 356, and grammar issue of “it add” in line 364. Please carefully review the entire text.
2. A direct pixel union operation between the outputs of SAM and raw binary facsimiles in Step-1 may introduce unrelated area to char-level masks? What I understand is to use raw binary facsimiles to fill in the missing pixels in the output of SAM. Any intermediate steps are omitted in the description?
3. Is there any reason for setting the iou threshold to 0.6?
4. How is the detection performance based on the YOLOv12 model? Will it introduce noise interference to the pipeline?
5. In Fig. 3, what is the relationship between the right image of "Expert-annotated Facsimiles in OBIMDTest" and the binaryized character-level image on the left? It seems that the correspondence between the low-quality part of the OBIMD raw and the binary image is inconsistent.
6. Lack of implementation details for training SAM2 （Sec. 4.1）
7. Lack of comparison with other segmentation models (Tab. 1), except the SAM series.
8. Has there been any comparison regarding the ground-truth? At present, it seems that the results of OBI CHARiot in Fig. 4 are somewhat different from those of other models and the raw data in terms of their outer contour.
9. At present, the qualitative results are a bit limited. It is suggested to add more case visualizations.
10. Lacking quantitative indicators such as PSNR, SSIM and other image generation metrics, the outputs of the denoising model and the OBICHARiot model were compared to provide numerical supplements as shown in Figure 5.
11. Only one denoising model Segformer has been compared*.* It is suggested to include the denoising models released in the past two years (e.g. CharFormer, ) as well as the models specifically designed for Oracle denoising.

---

> ### Author Response · Authors · 2025-11-29
>
> Thank you for your professional feedback.
>
> **Q1: The technical contributions are limited**
>
> **There are fundamental differences between our approach and SAM, extending beyond mere engineering adjustments.**
> SAM aims to progressively expand data starting from high-quality annotations, but **its data flywheel is not designed to handle pixel-level misalignment** between existing masks (i.e., facscimiles) and input images (i.e., rubbings) in OBIMD. In contrast, **OBI CHARiot is specifically designed to address this unique data quality challenge**.
>
> The alignment step in OBI CHARiot leverages dual SAMs by combining the character locations predicted by SAM-rub on the rubbings with the character masks extracted by SAM-fac from the facsimiles, thereby generating aligned annotations for the training set. This automatic alignment process effectively leverages misaligned facsimiles in OBIMD. In contrast, SAM’s iterative process adds the model’s high-confidence predictions to the training set directly. The two approaches differ fundamentally. Experimental results in Section 4 also demonstrate that our method significantly outperforms SAM in the context of rubbing segmentation.
>
> **Q2: reason for setting the iou threshold to 0.6?**
>
> We have added experiments in Section 4.6 regarding the IoU threshold in Stage 1. As detailed in Section 4.6, our ablation studies on the IoU threshold in Stage 1 reveal that IoU=0.6 offers the best data efficiency in Step 2, while the method remains robust to a range of IoU thresholds in Step 3.
>
> **Q3: Performance of YOLO? Will it introduce noise?**
>
> ||precision|recall|Box AP50|Box AP|
> |-|-|-|-|-|
> |Yolo|91.40|87.92|93.12|67.33|
>
> |Prompt|mask AP50|mask AP|miou|fscore|
> |-|-|-|-|-|
> |gt bbox|91.38|39.46|68.17|80.95|
> |yolo bbox|73.78|30.13|64.55|78.07|
>
> Compared to using gt bbox, there remains a noticeable performance gap when using YOLO-predicted bounding boxes. However, results in Section 4.3 show that the data annotated by OBI CHARiot-2 using YOLO boxes still enables the instance segmentation model to achieve a 6.14% improvement in mask AP50.
>
> Detailed ablation studies along with visualizations have been added in Section 4.6.
>
> **Q4: More experiments**
>
> **[Image generation metrics]**
> We have supplemented the evaluation with PSNR and SSIM metrics. OBI CHARiot achieves higher image quality. The results can be found in Table 4.
>
> **[More character denoising model]**
> We have added comparisons with CharFormer [Shi et al., 2022] across all experiments in Sections 4.2 to 4.5. Detailed results can be found in the paper.
>
> **Q5: Other detailed questions**
>
> **Prompt for SAM2**: In OBI CHARiot-1, gt bboxes are used as prompts during the alignment step, while in OBI CHARiot-2, YOLO-predicted bboxes are used as prompts, as mentioned in the paper.
>
> **Pixel union operation**: *This was a wording error: it should be "intersection" instead of "union."* In Step 1, we observe that while HQ-SAM's output lacks structural clarity, it effectively removes non-character noise. In contrast, the original facsimile retains better character structure. Therefore, taking the intersection of the two helps complement each other's strengths. Please refer to Step 1 in Figure 2 for a visual illustration.
>
> **Illustration in Fig. 2 should connect two stages**: The figure has been updated to include content related to Stage 2.
>
> **Information about experts**:This has been described in the "Test Set Construction" part of Section 4.1.
>
> **Decipherment is a classification task**:As mentioned in Section 2.1, existing OBI decipherment studies often follow different paradigms, such as character classification, character generation, and VLM interpretation. Among these, using a classification-based approach for decipherment can more intuitively demonstrate that the characters segmented by OBI CHARiot exhibit better glyph quality.
>
> **Relationship between images in Fig. 3 (now Fig. 4)**:The left column shows the facsimiles, and the right column shows the char-level masks. In OBIMD, the masks are obtained by directly extracting the foreground regions within the gt bboxes.
>
> **Implementation details for training SAM2**: We train SAM2 based on the original configuration without modifying key parameters. The configuration files will be made publicly available along with the code.
>
> **Lack of comparison with other segmentation models**: SAM2 is currently the most widely used model in segmentation data flywheel frameworks. We believe that the results obtained with SAM2 are sufficiently representative. Additionally, in Section 4.3, we compare OBI CHARiot with more instance segmentation models, and it still achieves superior performance.
>
> **OBI CHARiot in Fig. 4 (now Fig. 5) are somewhat different**:In Step 1 of OBI CHARiot, non-character noise is filtered out. In Step 3, the masks are extracted by SAM-fac rather than being directly extracted using gt bboxes. These processing steps may lead to slight contour differences in the final output.

---

### Official Review · Reviewer_CW21 · 2025-10-30

**Soundness:** 2
**Presentation:** 2
**Contribution:** 2
**Rating:** 4
**Confidence:** 4

**Summary:**

This paper addresses the task of stroke-level segmentation in oracle bone script. The authors observe that previous segmentation models typically take single-character images as input, thereby failing to effectively leverage background information that might improve segmentation. They instead propose feeding page-level images into the segmentation model.

However, prior work lacked precise, page-level annotations, only coarse annotations were available. To address this, the authors design a data annotation pipeline similar to the one in Segment Anything Model (SAM): starting from coarse annotations, they train a model, use the trained model to generate slightly more refined labels for the training data, and then iteratively refine both the model and its pseudo-labels to achieve increasingly accurate page-level annotations. The iteratively optimized annotations, according to the authors, yield models that outperform those trained on the original coarse labels. In addition, the authors manually refine the coarse annotations of the test set to create high-quality ground truth.

**Strengths:**

1. This paper proposes replacing single-character inputs with page-level inputs, enabling the model to utilize background cues, which improves segmentation performance.
2. This paper adapts a SAM-style automatic annotation pipeline to the oracle bone script segmentation task, using iterative pseudo-supervised training to improve annotation accuracy; annotations refined in this way yield slightly better model performance compared to the original coarse annotations.
3. This paper manually curated precise annotations for the test set and plan to release them publicly.

**Weaknesses:**

1. The idea of using page-level images to leverage background information is straightforward. The proposed automatic annotation pipeline is essentially the same as the pseudo-supervised iterative training described in the SAM paper, with only minor engineering adjustments for oracle bone script segmentation. The work is largely an engineering application with minimal conceptual novelty.

2. The paper claims that the optimized SAM-based pipeline produces better annotations than the original SAM pipeline, but does not clearly explain the specific implementation differences between the two. A clear side-by-side comparison (in either figures or text) is needed to highlight how they differ.

3. In lines 361–366, the analysis attributes the performance gap between the original SAM pipeline and the authors’ optimized version to the latter’s ability to use existing coarse bounding boxes to filter out inaccurate predictions. However, the original SAM pipeline did not use this information simply because its target images were unlabelled Internet data without any coarse boxes, not because this use of box information was a non-obvious idea. When coarse bounding boxes are available, filtering with them is straightforward and should not be considered a principal innovation.

4. According to Figure 4, the manual refinement of the test set annotations mainly involved removing distracting shell-edge background artifacts. Given that the dataset already has character bounding-box annotations, this could be achieved trivially by zeroing out all pixels outside the boxes. Thus, Figure 4 suggests that these background artifacts could be eliminated by a simple rule without the need for manual intervention. While manual refinement is of course more precise than automatic processing, the paper does not convincingly demonstrate that the improvement over simple automatic filtering is significant or necessary.

**Questions:**

In Figure 2, the IoU threshold for the alignment step in the automatic annotation pipeline is set to 0.6. How sensitive is the result to this choice? What impact would using different IoU thresholds have?

---

> ### Author Response · Authors · 2025-11-29
>
> Thank you for your professional feedback.
>
> **Q1: How sensitive is the model's performance to the IoU threshold?**
> We have added experiments in Section 4.6 regarding the IoU threshold in Stage 1.
>
> **[Step 2]** High-quality characters are selected by comparing the IoU between the binarized template and char-level masks.
>
> |Iou threshold|OBC samples|mask AP|mask AP50|
> |-|-|-|-|
> |0(step1 data)|64729|25.83|84.02|
> |0.5|12475|35.26|90.32|
> |0.55|7168|35.30|89.65|
> |0.6|3526|35.57|89.52|
> |0.65|1447|33.69|87.33|
> |0.7|426|29.19|83.19|
>
> The results show that IoU=0.6 is more data-efficient. Increasing the number of samples does not significantly improve performance, and using the full dataset even leads to a clear decline. However, reducing the sample size also causes a noticeable drop in performance.
>
> **[Step 3]** In the alignment step, OBC with IoU greater than the threshold from SAM-rub and SAM-fac are added to the subsequent training set.
>
> |Iou threshold|Num Iteration|mask AP|mask AP50|
> |-|-|-|-|
> |0|0|25.83|84.02|
> |0.5|2|38.65|91.00|
> |0.55|3|38.75|90.91|
> |0.6|4|39.46|91.38|
> |0.65|4|39.54|91.00|
> |0.7|4|39.44|91.06|
>
> The results indicate that the method is robust to this parameter. Setting the IoU threshold between 0.5 and 0.7 leads to only minor differences in final model performance.
>
> **Q2: Is the proposed pipeline essentially the same as SAM?**
>
> **There are fundamental differences between our approach and SAM’s methodology, extending beyond mere engineering adjustments.**
> SAM aims to progressively expand data starting from high-quality annotations, but **its data flywheel is not designed to handle pixel-level misalignment** between existing masks (i.e., facscimiles) and input images (i.e., rubbings) in OBIMD. In contrast, **OBI CHARiot is specifically designed to address this unique data quality challenge**.
>
> The alignment step in OBI CHARiot leverages dual SAMs by combining the character locations predicted by SAM-rub on the rubbings with the character masks extracted by SAM-fac from the facsimiles, thereby generating aligned annotations for the training set. This automatic alignment process effectively leverages misaligned facsimiles in OBIMD. In contrast, SAM’s iterative process adds the model’s high-confidence predictions to the training set directly. The two approaches differ fundamentally. Experimental results in Section 4 also demonstrate that our method significantly outperforms SAM in the context of rubbing segmentation.
>
> Regarding your suggestion that the performance gap stems from OBI CHARiot’s 'use of existing coarse bounding boxes to filter out inaccurate predictions', this interpretation requires clarification.
>
> **The main limitation of SAM’s iterative process lies in its inability to fully utilize the original misaligned ground-truth facsimiles to identify challenging samples.** As a result, SAM struggles to accurately segment difficult characters or complex structures. Our experiments show that SAM sometimes adds high-confidence yet structurally flawed predictions to the training set. Consequently, as the model continues to favor these easier-to-segment examples, the refinement of character structures stagnates in subsequent training iterations. In contrast, OBI CHARiot, through alignment rather than self-training, effectively exploits the low-quality data in OBIMD, leading to more accurate segmentation of complex structures. Our visualization (Figure 5 and Appendix B) also confirm that our method produces clearer and more complete structures compared to SAM.
>
> **Q3: Could the test set be trivially annotated by zeroing out all pixels outside the boxes?**
>
> In the newly added Figure 3, we highlight three major issues in OBIMD. While pre-trained HQ-SAM can effectively remove most non-character information and adjacent-character noise with box prompt, the core challenge in OBIMD remains the widespread pixel-level misalignment between the foreground pixels in rubbing images and their corresponding facsimiles.
>
> In fact, the "raw data" column in Figure 5 and Appendix B shows that the character masks obtained by simply zeroing out pixels outside the bounding boxes are of poor quality. Therefore, to enable reliable evaluation, we engage domain experts to perform manual annotations.

---

### Official Review · Reviewer_J7Bw · 2025-10-31

**Soundness:** 3
**Presentation:** 3
**Contribution:** 2
**Rating:** 4
**Confidence:** 4

**Summary:**

Aiming at the defects of traditional paradigms and dataset issues in the field of Oracle Bone Inscription (OBI) rubbing segmentation, this paper proposes a new full-page segmentation paradigm and a two-stage OBI CHARiot framework. It optimizes data quality and scale through a data flywheel mechanism, constructs a high-quality test set, and experimental verification shows that it has significant advantages in segmentation performance and downstream tasks. Additionally, the publicly available resources facilitate research in this field. Overall, the design is reasonable, with strong innovation and practicality.

**Strengths:**

1. It breaks through the traditional "character-level denoising" paradigm for Oracle Bone Inscription (OBI) rubbings and proposes a new "full-page rubbing segmentation" paradigm to synergistically preserve character structures and full-page contextual information, filling the gap in this field.
2. Additionally, it constructs the expert-annotated OBIMDTest dataset, subdivides evaluation scenarios, and innovates the evaluation system.
The research quality is solid: the method features strong implementability and targeted handling of rubbing noise, while the experimental design is reliable with multi-dimensional comparative verification.
3. The expression is highly clear: the technical details are explicit and supplemented with figures and tables, and the information transparency is high.
4. Academically, it provides a transferable framework for processing low-quality and sparsely labeled data in specific domains, filling relevant research gaps, and offering benchmark references. In terms of application, it aligns with the practical needs of OBI decipherment, the generated outcomes can assist experts in their work, and the publicly released resources promote the improvement of digital research efficiency and technical standardization in this field.

**Weaknesses:**

1. The paper mentions achieving iterative alignment through SAM-rub (for processing rubbings) and SAM-fac (for processing facsimiles), yet it fails to clarify the feature interaction mechanism between the two models (e.g., whether they share feature layers, how to resolve alignment biases caused by modal differences) and does not explain how edge distortion is avoided during the mask resolution enhancement process
2. The parameters of the data flywheel lack optimization verification: the impact of iteration count and IoU threshold (currently set to 0.6) on performance has not been explored, making it impossible to determine the optimal configuration and convergence conditions. It is recommended to supplement comparative experiments under different parameters and provide the basis for parameter selection
3. In the second stage, YOLOv12 is used to generate character prompt boxes for SAM, but the paper does not explain the impact of YOLOv12's detection accuracy on SAM's segmentation results (e.g., how missed detection/false detection boxes are handled) nor clarify whether a prompt box correction mechanism is designed. This may cast doubt on the reliability of automatic annotation for unlabeled data
4. The coverage of comparative models is insufficient: models suitable for low-data/small-sample segmentation scenarios and lightweight Transformer-based models are not included, and only ResNet50 is used as the classifier in downstream tasks. The demonstration of compatibility and universality is inadequate. It is recommended to supplement comparative experiments with such models to verify the framework's advantages
5. Details on noise handling are lacking: the paper does not elaborate on how to distinguish between character noise and non-character interference (e.g., boundary lines) nor verify noise resistance. It is recommended to supplement technical details of noise handling and experimental results on high-noise subsets
6. The zero-shot verification scenario is single: only HuaDong rubbings are used for evaluation, without covering rubbings from different periods or produced via different processes, making it impossible to demonstrate broad generalization ability. It is recommended to supplement zero-shot test sets from multiple sources and analyze the causes of performance degradation

**Questions:**

1. For the dual SAMs (SAM-rub and SAM-fac), the feature interaction mechanism and the specific structure of the deconvolution layer are not clarified, nor is it explained how edge distortion is avoided during resolution enhancement. Could you supplement the schematic diagram of model interaction and the edge fidelity solution?
2. In the second stage, the detection accuracy (miss rate, false detection rate) of YOLOv12 and the impact of detection errors on SAM segmentation are not mentioned. Could you provide the mAP index of YOLOv12 and the prompt box correction strategy (if any)?
3. For the termination of the data flywheel iteration, only "almost no additional aligned data" is mentioned, with no quantitative standard. Could you specify the termination index (e.g., the proportion of new additions in N consecutive rounds < X%) and the basis for its selection?
4. Whether a multi-task loss function is used for full-page facsimile and character mask generation, and how the loss weights are allocated, are not explained. Could you supplement the design of the loss function and the impact of weights on the performance of the two tasks?

---

> ### Author Response · Authors · 2025-11-29
>
> Thank you for your professional feedback.
>
> **Q1: Clarification on the dual SAMs design**
>
> SAM-rub and SAM-fac are two independent models with no feature-level interaction between them. Regarding the deconvolution layers, we employ a two-layer ConvTranspose2d to increase resolution without altering the number of channels.
>
> **How is edge distortion avoided?** SAM2 takes a 1024×1024 input and outputs a mask of 256×256, which is then upsampled to 1024×1024 using nearest-neighbor interpolation. Previous works (e.g., HQ-SAM [Ke et al., 2023] and Hi-SAM [Ye et al., 2024]) have attempted to improve mask quality by replacing nearest-neighbor upsampling with lightweight deconvolution layers, demonstrating that this approach yields higher-quality masks. Our incorporation of deconvolution layers is inspired by these studies. For Oracle Bone Characters  (OBC) segmentation, character edges play a critical role in preserving structural integrity, and we plan to further explore this in future work.
>
> **Q2: Detection accuracy of YOLOv12 and the impact of detection errors**
>
> ||precision|recall|Box AP50|Box AP|
> |-|-|-|-|-|
> |Yolo|91.40|87.92|93.12|67.33|
>
> |Prompt|mask AP50|mask AP|miou|fscore|
> |-|-|-|-|-|
> |gt bbox|91.38|39.46|68.17|80.95|
> |yolo bbox|73.78|30.13|64.55|78.07|
> |grid points(32x32)|63.16|23.65|49.44|64.65|
>
> Compared to using gt bbox, there remains a noticeable performance gap when using YOLO-predicted bounding boxes. However, experimental results in Section 4.3 show that the data automatically annotated by OBI CHARiot-2 using YOLO boxes still enables the instance segmentation model to achieve a 6.14% improvement in mask AP50.
>
> **Is there a box correction strategy?**  We use the bounding box of the mask predicted by SAM2 based on the YOLO box as the final box for training.
>
> Additionally, in Section 4.6, we have included ablation studies related to YOLO and SAM’s "segment everything" mode, along with corresponding visualization results.
>
> **Q3: Parameters of the data flywheel lack optimization verification**
>
> **[IoU threshold]**
> We have added experiments in Section 4.6 regarding the IoU threshold in Stage 1.
>
> **[Step 2]**
> |Iou threshold|OBC samples|mask AP|mask AP50|
> |-|-|-|-|
> |0(step1 data)|64729|25.83|84.02|
> |0.5|12475|35.26|90.32|
> |0.55|7168|35.30|89.65|
> |0.6|3526|35.57|89.52|
> |0.65|1447|33.69|87.33|
> |0.7|426|29.19|83.19|
>
> The results show that IoU=0.6 is more data-efficient.
>
> **[Step 3]**
> |Iou threshold|Num Iteration|mask AP|mask AP50|
> |-|-|-|-|
> |0|0|25.83|84.02|
> |0.5|2|38.65|91.00|
> |0.55|3|38.75|90.91|
> |0.6|4|39.46|91.38|
> |0.65|4|39.54|91.00|
> |0.7|4|39.44|91.06|
>
> The results indicate that the method is robust to this parameter.
>
> **[Iteration counts]**
> The iteration continues until fewer than 1,000 (in Stage 1) or 5,000 (in Stage 2) new OBCs are aligned in a single step. This stopping criterion has been clearly stated in the paper. Additionally, we have provided the detailed number of iterations under different IoU thresholds for OBI CHARiot-1 in Section 4.6.
>
> **Q4: The coverage of comparative models is insufficient**
> In Section 4.3, we have added comparisons with the lightweight Transformer-based model **Mask2Former**. In Section 4.5, we further included convolutional-based **EfficientNet** and Transformer-based **ViT and Swin Transformer**. Experimental results (**see Table 2 and Table 4**）demonstrate that OBI CHARiot maintains a performance advantage over these models.
>
> **Q5: Whether a multi-task loss function is used?**
> We merge all character masks detected on a single rubbing to reconstruct the full-page facsimile. During training, the model is only optimized for the character mask generation task.
>
> **Q6: Details on noise handling are lacking**
> There may be a misunderstanding here. OBI CHARiot addresses noise present in the original facsimiles, not noise in the input rubbings. The robustness of OBI CHARiot mainly reflects its out-of-distribution (OOD) generalization capability, rather than robustness to input-level noise.
>
> In the newly added Figure 3, we illustrate typical noise types in OBIMD, including adjacent character, pixel misalignment and non-character noise. These are handled step-by-step in OBI CHARiot-1:
>
> - For non-character information and adjacent character, we observe that using the pre-trained HQ-SAM with the facsimile as input can effectively filter such noise (see Step 1 in Figure 2).
> - For pixel misalignment, we address it iteratively through the data flywheel process in Step 3.
>
> **Q7: Zero-shot scenario is single**
> Due to data limitations (OBIMD currently only contains rubbings from HeJi and HuaDong), zero-shot evaluation is only performed on the HuaDong set. In the future, we plan to extend the dataset to include rubbings from other historical periods. Additionally, we have included a performance analysis on the OBIMDTest-HD set in Appendix C.

---

### Official Review · Reviewer_Zgy7 · 2025-11-01

**Soundness:** 2
**Presentation:** 2
**Contribution:** 3
**Rating:** 4
**Confidence:** 4

**Summary:**

This paper proposes OBI CHARiot, a two-stage framework for full-page Oracle Bone Inscription (OBI) rubbing segmentation. It introduces a new paradigm that captures both character structure and contextual information. The first stage aligns misaligned rubbing–facsimile pairs through an iterative data flywheel process, while the second expands the dataset using automated annotation of unlabelled rubbings. A new expert-annotated test set, OBIMDTest, is also built. Experiments show large gains in segmentation accuracy and downstream OBI deciphering, demonstrating the framework’s effectiveness in improving both model performance and data quality.

**Strengths:**

1. The paper accurately identifies the core problem with the existing OBI dataset (OBIMD)—namely, the pixel-level misalignment between facsimiles and rubbings—and the limitation of the existing paradigm (character-level denoising) in losing context.

2. The OBI CHARiot framework is a clever, multi-stage solution. The Stage 1 data flywheel for correcting misalignment and the Stage 2 data flywheel for expanding the dataset is a "fix-then-enhance" strategy that is highly logical and effective.

3. The authors invested the effort to create an expert-annotated test set (OBIMDTest). This greatly enhances the credibility of the experimental results, as the evaluation no longer depends on the low-quality labels of the original OBIMD dataset.

**Weaknesses:**

1. In Stage 2, the authors use a YOLOv12 model to generate bounding boxes as prompts for SAM. The justification is that OBI characters are too sparse for SAM's default grid-based prompting to work. This is a strong claim, but the paper does not seem to provide experimental data (e.g., the recall of SAM's "segment everything" mode) to fully support this choice. This adds an extra model dependency.

2. The paper fails to specify the exact number of iterations for the data flywheel mechanism, and the selection of hyperparameters such as IoU appears rather arbitrary. The authors should provide a more rational explanation and conduct experiments to validate the choices of these hyperparameters.

**Questions:**

1. The paper mentions using SAM2 (Ravi et al., 2024) in Section 4.1 (Compared Training Frameworks), but primarily cites SAM (Kirillov et al., 2023) in the methodology (Section 3). Was the OBI CHARiot framework built on SAM or SAM2? Please clarify.

2. The use of YOLOv12 in Stage 2 is justified by SAM's grid-prompting failing due to pixel sparsity. Could you provide a brief ablation or data to support this? For instance, what is the recall of SAM's "segment everything" mode on the OBIMDTest set?

3. How many iterations did the data flywheels in Stage 1 and Stage 2 each require to converge? Furthermore, the 0.6 IoU threshold for alignment in Stage 1 is a hyperparameter; how sensitive is the model's performance to this threshold?

---

> ### Author Response · Authors · 2025-11-29
>
> Thank you for your professional feedback.
>
> **Q1: Is the OBI CHARiot framework built on SAM or SAM2?**
> Our data flywheel draws inspiration from SAM’s methodology, while the model itself is based on SAM2. We have clarified this point further in the paper.
>
> **Q2: Ablation on SAM's "segment everything" mode?**
> We have supplemented experiments in Section 4.6 and provided visualization results.
>
> |Prompt|mask AP50|mask AP|miou|fscore|
> |-|-|-|-|-|
> |gt bbox|91.38|39.46|68.17|80.95|
> |yolo bbox|73.78|30.13|64.55|78.07|
> |grid points (32x32)|63.16|23.65|49.44|64.65|
>
> Compared to YOLO boxes, using grid points performs worse on OBJMDTest-HJ. Visualization results in Section 4.6 indicate that this is mainly because grid points are distributed more densely relative to Oracle Bone Characters(OBC). Many points fall outside character regions, leading to potential false positives when hitting noise (e.g., cracks or abrasions). Additionally, a single OBC containing multiple radicals may be oversegmented into multiple characters. Therefore, we used YOLO boxes in OBC CHARiot-2.
>
> **Q3: How sensitive is the model's performance to the IoU threshold?**
> We have added experiments in Section 4.6 regarding the IoU threshold in Stage 1.
>
> **[Step 2]** High-quality characters are selected by comparing the IoU between the binarized template and char-level masks.
>
> |Iou threshold|OBC samples|mask AP|mask AP50|
> |-|-|-|-|
> |0(step1 data)|64729|25.83|84.02|
> |0.5|12475|35.26|90.32|
> |0.55|7168|35.30|89.65|
> |0.6|3526|35.57|89.52|
> |0.65|1447|33.69|87.33|
> |0.7|426|29.19|83.19|
>
> The results show that IoU=0.6 is more data-efficient. Increasing the number of samples does not significantly improve performance, and using the full dataset even leads to a clear decline. However, reducing the sample size also causes a noticeable drop in performance.
>
> **[Step 3]** In the alignment step, OBC with IoU greater than the threshold from SAM-rub and SAM-fac are added to the subsequent training set.
>
> |Iou threshold|Num Iteration|mask AP|mask AP50|
> |-|-|-|-|
> |0(step1 data)|0|25.83|84.02|
> |0.5|2|38.65|91.00|
> |0.55|3|38.75|90.91|
> |0.6|4|39.46|91.38|
> |0.65|4|39.54|91.00|
> |0.7|4|39.44|91.06|
>
> The results indicate that the method is robust to this parameter. Setting the IoU threshold between 0.5 and 0.7 leads to only minor differences in final model performance.

---

### Author Response · Authors · 2025-11-30
**Brief Summary of Rebuttal Stage**

Dear ACs and Reviewers,

We appreciate your valuable time on our work. Below is a summary of our work and our responses during the rebuttal stage.

**Task Definition**: Taking OBI rubbing images as input, the task aims to generate facsimiles (i.e., binarized masks) and character-level segmentation results.

**Existing Issues**: Existing method mainly denoise character regions, directly segmenting OBI characters from full-page rubbings remains unexplored, primarily due to the scarcity of high-quality annotated data.

**Our Contributions**:

1. Paradigm: Rather than the Character-level Denoising Paradigm adopted by most existing methods, which ignores the contextual information, we propose a novel Full-page Rubbing Segment Paradigm that achieves superior performance on this task.
2. Framework:  To address the issues of low-quality and scarcity in public datasets, we present OBI CHARiot, a two-stage training framework based on data flywheel mechanism. OBI CHARiot fully leverages existing misaligned ground-truth through iterative character alignment and expands the dataset volume by automatically annotating unlabeled rubbings.
3. Data: Data processed by OBI CHARiot leads to superior model performance. Specifically, various instance segmentation models trained on our processed data outperform those trained on the raw data by 22.37% in mask AP50.

**Summary of Rebuttal**

The reviewers highly recognize our contributions, believing that OBI CHARiot successfully adapts the automatic annotation process of SAM to the task of rubbing segmentation, and the results of experiments demonstrate its stable performance advantages.

On this basis, the reviewers also raised some questions about our work. We have conducted in-depth experiments targeting these questions and updated the results in our paper. Herein, we have compiled the most common questions for your reference.

**Q1: What are the key differences between the iterative process of OBI CHARiot and that of SAM?**

**There are fundamental differences between our approach and SAM’s methodology, extending beyond mere engineering adjustments.**
SAM aims to progressively expand data starting from high-quality annotations, but **its data flywheel is not designed to handle pixel-level misalignment** between existing masks (i.e., facscimiles) and input images (i.e., rubbings) in OBIMD. In contrast, **OBI CHARiot is specifically designed to address this unique data quality challenge**.

The alignment process in OBI CHARiot leverages dual SAMs by combining the character locations predicted by SAM-rub on the rubbings with the character masks extracted by SAM-fac from the facsimiles. This design effectively address the misalignments in the OBIMD dataset. In contrast, SAM’s iterative process adds the model’s high-confidence predictions to the training set directly. Experimental results also demonstrate that our method significantly outperforms SAM in the context of rubbing segmentation.

**Q2: How to select IoU threshold? Is OBI CHARiot sensitive to IoU threshold?**

We have added experiments in Section 4.6 regarding the IoU threshold in Stage 1.

Our ablation studies on the IoU threshold in Stage 1 reveal that IoU=0.6 offers the best data efficiency in Step 2, while the method remains robust to a range of IoU thresholds in Step 3.

**Q3: Does using YOLO boxes in OBI CHARiot-2 offer advantages over grid points?**

|Prompt|mask AP50|mask AP|miou|fscore|
|-|-|-|-|-|
|yolo bbox|73.78|30.13|64.55|78.07|
|grid points (32x32)|63.16|23.65|49.44|64.65|

We have added ablation studies and visualizations in Section 4.6.

Compared to YOLO boxes, using grid points yielded inferior results on OBIMDTest-HJ and is less efficient (~25x slower) due to dense point sampling. The core issue is that the default grid-point sampling, tailored for panoptic segmentation, may leads to false positives (when points hit cracks/abrasions), false negatives (missing small characters with no sampled points), and over-segmentation of compound characters. Consequently, OBI CHARiot-2 utilizes YOLO boxes.

---

Additionally, the following content has been added due to reviewers concerns:

1. **Comparison with CharFormer (Sec 4.2-4.5):** Implementation of the character-denoising paradigm using the recent CharFormer [Shi et al., 2022] method. Results show our OBI CHARiot outperforms it, reinforcing our paradigm's advantage.
2. **Image Quality and Classifier (Sec 4.5):** Evidence from SSIM/PSNR and an additional classifier confirming the superior character structure segmented by OBI CHARiot.
3. **Extended Visualizations (Appendix B & C):** Additional qualitative results and analysis for clearer illustration.

We hope that our responses and the revisions made to the paper addressed the reviewers' questions and concerns.  Once again, we would like to express our gratitude to the  ACs and reviewers for their constructive suggestions, which have greatly helped improve the quality of our work.

Sincerely,

The Authors

---

### Meta-Review · Area_Chair_gx7N · 2026-01-06

**Summary:**

The reviewers generally recognize the contribution to the OBI community and note the empirical gains for the segmentation and decipherment tasks.
The primary concerns center on limited conceptual novelty and contributions, with multiple reviewers characterizing the method as largely an engineering-oriented adaptation of existing SAM-style pseudo-supervised pipelines rather than a fundamentally new algorithmic contribution. As its application and engineering-oriented design, the proposed framework contains many components and practical designs. Additional concerns include insufficient clarity and justification of design choices (e.g., IoU thresholds, iteration stopping criteria, use of YOLO prompts), missing or unclear technical details in earlier versions, such as dual-SAM design and noise handling, and annotation mechanisms, and incomplete comparisons and limited datasets.

The authors’ rebuttal directly addresses most reviewer questions about the technical detail by providing additional experiments or implementation and experimental details. While the responses provide additional information that tends to mitigate concerns about the robustness and unclear setup in the implementation and experiments, the rebuttal does not fully resolve the broader concern regarding limited methodological novelty, which remains the main factor keeping the paper marginal. Additionally, while the rebuttal provides clarifications on the technical details raised by the reviewers, the underlying concerns regarding the sufficiency of design justifications are only partially resolved. Given these remaining and unaddressable issues, particularly relative to the level of the venue, the AC recommends rejection.

**Reviewer Concerns:**

Most reviewers’ questions regarding technical details are addressed by the rebuttal through the provision of additional information and experiments, including justifications for hyperparameter choices like IoU thresholds, iteration numbers, and stopping condition, the use of YOLO for prompt generation, and clarification of the use of SAM versus SAM2.

However, the primary concerns regarding limited conceptual novelty and contribution, restricted validation, and issues in presentation and overall clarity remain unresolved. While additional technical details are provided, these largely amount to direct or post hoc responses to specific questions and do not sufficiently address the underlying issues of weak design justification and the predominantly engineering-oriented nature of the proposed framework.

**Reviewer Scores:**

- Reviewer Zgy7.
Most technical and justification concerns were directly addressed with new experiments and clarifications. While specific design details were provided, the weakness on the dependency of the selection of these details may persist. The reviewer may maintain their original score, with some potentially leaning toward a slight improvement from 4 to 5.

- Reviewer J7Bw.
Detailed responses on parameters, YOLO usage, comparisons, and design clarity mitigate most concerns, though novelty remains moderate. The reviewer would likely maintain their original score, with some potentially leaning toward a slight improvement from 4 to 5.

- Reviewer CW21.
Questions on IoU threshold and pipeline distinctions from SAM were responded, but concerns about conceptual novelty and sensitivity to the design details may persist. The reviewer may maintain the original score 4.

- Reviewer Lf6A.
Many questions related to technical and clarity issues were responded, but concerns about limited novelty and framing of contributions likely remain. The reviewer may maintain the original score 4.

---

### Decision · Program_Chairs · 2026-01-26

Reject